

# A daily gridded high-resolution meteorological data set for historical impact studies in Switzerland since 1763

Noemi Imfeld[1,2] and Stefan Brönnimann[1,2]

[1]Oeschger Center for Climate Change Research, University of Bern, Switzerland
[2]Institute of Geography, University of Bern, Switzerland

**Correspondence:** Noemi Imfeld (noemi.imfeld@unibe.ch )

**Abstract.** High-resolution gridded daily data is needed to study historical climate and weather impacts. Current daily gridded data sets in Switzerland extend to 1961 or 1971 for variables such as minimum and maximum temperature and sunshine duration. However, studying historical weather and climate events, such as the year-without-a-summer requires much longer time periods. For Switzerland, high-resolution gridded reconstructions of daily mean temperature and daily precipitation sums have recently been developed based on a large amount of early instrumental data for a period from 1763 to 1960. Here, we present an extension of these daily reconstructions to six more variables, namely, relative sunshine duration, relative humidity, minimum and maximum temperature at 2 m, and u- and v-wind at 10 m with a 1x1 km resolution. These additional reconstructions are based on the same method as the previous reconstructions by combining the analogue resampling method and data assimilation. Cross-validation results using a network representative of early 19th-century observations show a mean squared error skill score ranging from 0.70 to 0.80 for wind speed, depending on the season. For maximum and minimum temperature, values average between 0.48 to 0.82, depending on the seasons. These results indicate reasonable skill of the reconstructions and show that the wind and temperature fields outperform climatology despite the data scarcity in the historical period. However, for relative humidity and relative sunshine duration, the values of the mean squared skill score are significantly lower, ranging between -0.31 to 0.48. Furthermore, we explored the potential of the extended reconstructions by evaluating historical and contemporary wildfire events in Switzerland using the widely used Canadian Forest Fire Weather Index (FWI). The two historical fires were associated with a notably high fire danger in the reconstruction. For the contemporary winter fire, the reconstruction agrees well with the index calculated from the COSMO-1 weather forecast model, though neither indicates exceptionally high fire danger.

Overall, this is the first data set that enables impact studies of weather and climate in Switzerland, reaching as far back as 1763.

## 1 Introduction

Long-term high-resolution reconstructions of meteorological variables are crucial for studying historical climate variability, understanding the occurrence of past extremes, and assessing their impacts on, for example, hydrology and agriculture. Over the past years, significant data rescue efforts uncovered numerous observational records for the 18th and 19th centuries that





are, however, sparse in spatial coverage (e.g., Brugnara et al., 2015, 2020). While gridded data sets for daily mean temperature, daily precipitation sums, or daily mean pressure have been reconstructed from such rescued records for Switzerland (Pfister et al., 2020; Imfeld et al., 2023) and Europe (Pappert et al., 2022; Schmutz et al., 2024), gridded data sets for other important variables — such as wind speed, humidity, sunshine duration, and minimum and maximum temperatures — are very scarce for long-term historical periods. For example, dynamical downscaling approaches have been used to study a variety of variables

for specific extreme events on hourly to daily scales (e.g. Stucki et al., 2015, 2024; Michaelis and Lackmann, 2013), but no continuous long-term daily gridded data exist except for the 20th century reanalysis with a rather coarse resolution (Slivinski et al., 2019). This is primarily due to the lack or limited availability of measurements, which often suffer from questionable representativeness or quality (e.g., wind measurements taken on church towers). Yet, these variables are crucial for assessing, for example, agricultural impacts, drought conditions, and ecosystem responses.

In this study, we extended the existing daily reconstruction of mean temperature and precipitation sums (Imfeld et al., 2023) to include the u- and v-wind component at 10 m, mean and minimum daily relative humidity, daily relative sunshine duration, and minimum and maximum temperatures at 2 m covering the period 1763 to 2020. To ensure consistency with the previous reconstruction, we used the same set of observations and adhered as closely as possible to the original methodology. Our primary focus was providing fields for 10 m wind and daily maximum and minimum temperature at 2 m, for which we

performed data assimilation on top of the analogue resampling. The other variables — mean and minimum relative humidity and daily relative sunshine duration — are included in the data set because they are frequently requested, particularly for applications like crop modeling or fire weather conditions, however, these variables are solely based on analogue resampling without additional data assimilation.

The reconstructed variables enable the calculation of indices such as the Canadian Forest Fire Weather Index, providing

insights into historical fire weather conditions. To demonstrate the utility of the data set, we used two historical forest fire events in Switzerland (late summers of 1911 and 1943), complemented with documentary sources, to showcase its potential for analyzing past events and their impacts, as well as a modern event in the winter of 2016 to compare it two a modern weather forecast model.

The paper is organized as follows: Section 2 presents the observations and gridded data sets used for the reconstruction.

In Sect. 3, we describe the methodology of the reconstruction, and in Sect. 4, we show the corresponding validation results. Section 5 shortly discusses the long-term consistency of the variables, and Sect. 6 shows a use case of the data sets. Conclusions are drawn in Sect. 7.

## 2  Data

### 2.1  Instrumental data

Our reconstruction method requires both historical measurements and corresponding present-day observations from the analogue reference period. For our new reconstructions, we largely relied on the same set of observational records detailed in depth



in the first article (Imfeld et al., 2023), but additionally included observations for daily minimum and maximum temperature for the period since 1864. Therefore, we provide here only a brief description of the observational records.

The reconstruction period can be divided into two periods, from 1763 to 1863 and from 1864 to 2020 (including the reference periods). For the period before 1864, we relied on early instrumental data rescued through various initiatives (Camuffo and Jones, 2002; Klein Tank et al., 2002; Füllemann et al., 2011; Brugnara et al., 2020; Pfister et al., 2019; Brugnara et al., 2022). These early instrumental data required more pre-processing because standardized measurement practices were not yet established at the time of recording, resulting in inherent limitations in the records. For the reconstructions, we only included time series that passed basic quality controls with a minimum record length of at least seven continuous years and few data gaps. In the early periods of the reconstruction, the network coverage was much sparser, and over time, considerable changes occurred. The changes in the network are depicted in Fig. 1a. At the start of the reconstruction period in 1763, around 11 to 12 series were available, which increased to around 30 series in the mid-19th century. To increase the number of observations in the early period, we also added time series from nearby locations in Italy and Germany. In total, the reconstruction before 1864 is based on 17 pressure series, 18 temperature series, and 6 precipitation and precipitation occurrence series. Because only very few precipitation measurements were available, we included precipitation occurrence derived from weather notes. All observations underwent standard quality control procedures of daily mean values as implemented in the R-package "dataresqc" (Brugnara et al., 2019), and we performed an additional spatial quality control following Estévez et al. (2018). Temperature and pressure series were homogenized with the paleo-reanalysis EKF400v2 (Valler et al., 2022) as a reference series using the penalized maximal t-test (Wang et al., 2007) and the penalized maximal F-test (Wang, 2008) for break point detection. For the four main cities in Switzerland, Bern, Zurich, Basel, and Geneva, the merged series from Brugnara et al. (2022) were used.

For the later period, starting in 1864, the Swiss National Weather Service MeteoSwiss provides a dense, high-quality, and homogeneous network for daily mean, maximum, and minimum temperature and daily precipitation sums, as well as observations for daily mean pressure for Switzerland (Begert et al., 2005, 2007; Füllemann et al., 2011). These observations form the basis for our reconstruction between 1864 and 1970/2015 and for the reference period between 1971/2016 to 2020 (the different reference periods will be detailed in Sect. 3.2). The temperature and precipitation observations meet a high-quality standard, including homogenization and required no additional processing. For the reconstruction of minimum and maximum, we used only series without large gaps in the records. For the wind reconstruction, we selected a subset of stations that showed the best validation results for the analog resampling method, and instead of daily precipitation sums, we only used daily precipitation occurrences. The respective networks for the temperature and wind reconstructions are displayed in Fig. 1 b and c. The daily pressure observations, however, also showed quality problems for the period after 1864, including climatic outliers and inhomogeneities. Therefore, we applied the same methods to the pressure data after 1864, as mentioned above for the early instrumental data. We performed a more profound quality control using the dataresqc R-package (e.g., test for daily repetition, climatic outliers, duplicates, internal consistency) and a spatial quality control comparing estimated pressure to the measured values nearby as done in Estévez et al. (2018). We re-digitised 361 values where we had access to the respective variables from the "Wetterarchive" of MeteoSwiss (e.g., MeteoSwiss, 1868), we used the estimates from the spatial quality control of 190 values instead of the original values to not lose the data for some stations, and we in addition flagged 38 values as climatic out-

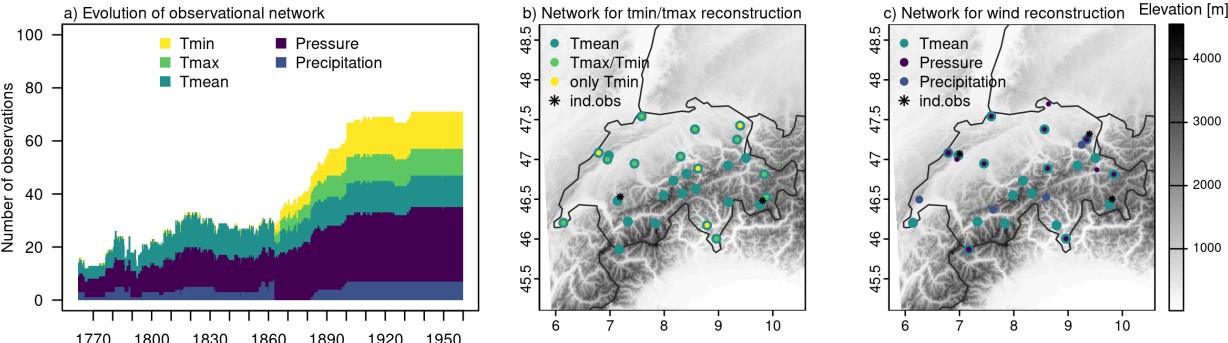

**Figure 1.** a) Evolution of available observations over time. b) Network for the reconstruction of maximum and minimum temperature for the period since 1864. c) Network for the reconstruction of wind fields for the periods since 1864. The networks prior to 1864 are identical to the data used in Imfeld et al. (2023) (see Fig. 1 therein). A black star indicates that measurements from this station have been used for an independent evaluation.

liers. If a value was flagged as a daily repetition, we kept it in the data set. We homogenized the quality-controlled series using surface pressure from the closest grid point of the ModERA reanalysis (Valler et al., 2024) as a reference series. The break point detection was performed with the penalized maximal t-test (Wang et al., 2007) and the penalized maximal F-test (Wang, 2008). Only break points that were significant without metadata were considered. This simple procedure led to more homogeneous wind reconstructions. We believe that a high-level quality control, involving the use of original documents to verify outliers, identify potential re-digitisation needs, and compile metadata for homogenisation, could lead to further improvements in the pressure time series. This was, however, beyond the scope of this study.

Each historical observation requires a corresponding counterpart in the reference period. If no reference station was available at the location of a historical station, we used the closest grid point from the daily temperature and precipitation fields or, for sea level pressure, from the European EOBS data set v23.1 (Cornes et al., 2018) during the reference period. For time series in Germany and Italy, if available, we used daily ECA&D station data (Klein Tank et al., 2002), and otherwise the closest grid points from EOBS. All reference series were gap-filled with a quantile mapping approach between the series and a gridded data set (mainly EOBS) following Gudmundsson et al. (2012) and were homogenised if this has not been done yet with nearby stations.

To independently evaluate the temperature reconstruction, we used the two homogeneous minimum and maximum temperature records from Segl Maria and Col du Grand St-Bernard, which were not included in the reconstruction process because they had large gaps in the early 20th century (see asterisks in Fig. 1b). For the wind reconstruction, we used wind speed measurements from three stations, Saentis, Segl Maria, and Neuchâtel, located in different areas of Switzerland (see asterisks in Fig. 1c). The other variables were not evaluated with independent observations, but we qualitatively assessed their long-term evolution.





## 2.2 Gridded data sets

The analogue fields for daily minimum and maximum temperature and daily relative sunshine duration were resampled from three daily spatial data sets from MeteoSwiss for a period from 1st January 1971 to 31st December 2020, which are available at a 1x1 km grid resolution. The daily minimum and maximum temperature fields are constructed using non-Euclidean distance weighting of the difference between the daily extremes and the daily mean temperature, ensuring that tmin < tmean < tmax (MeteoSwiss, 2021c). Daily relative sunshine duration (SrelD) is available in percent (%) from midnight to midnight, representing the ratio between the effective sunshine duration and the maximum possible sunshine duration when no clouds would be present (MeteoSwiss, 2021b). The spatial fields are predicted from approximately 70 in-situ measurements based on a kriging model with external drift and using the first nine Principal Components of nine years of satellite data as predictors. The evaluations of this data set show that characteristic radiation features of Switzerland, such as low-level stratus over the Swiss Plateau and Foehn, are realistically represented in the data set.

Fields of the u- and v-wind components at 10 m and relative humidity fields at 2 m were resampled from the analyses of the former weather forecasting model COSMO-1 from MeteoSwiss, which is available from March 2016 to October 2020, resulting in around four years of data. COSMO-1 is a nonhydrostatic deterministic limited-area numeric weather prediction model, which has been run at a 1.1. km grid over Switzerland (for further descriptions see Kruyt et al., 2018; Miralles et al., 2022). We also tested a resampling from downscaled wind fields from COSMO-1 between 1961 and 2020 using ERA-5 variables as predictors (Miralles et al., 2022). However, using these downscaled fields compared to the original COSMO-1 fields resulted in worse reconstructions for the entire period despite the larger analogue pool. We resampled the COSMO-1 field using bilinear interpolation to make it comparable to the 1x1 km grids (Swiss coordinate system LV95) of the other variables. The daily mean values were calculated from hourly data from midnight to midnight, and the minimum relative humidity was determined by extracting the lowest value recorded within the 24 hours from midnight to midnight.

## 2.3 Additional data sets

We restricted the selection of analogue days to days of similar synoptic weather situations using a new weather type reconstruction by Pfister et al. (2024) and not the old version of Schwander et al. (2017) as used in the previous reconstruction. Especially for the wind fields, an accurate representation of the weather types is relevant because they are used to calculate the background error covariance matrix (see Sect. 3.3). Due to the small reference sample size in the gridded data for the wind fields, we still perform the same merging of weather types as in Schwander et al. (2017) by combining weather types 5 and 8 (both high-pressure convective weather types), and 7 and 9 (both cyclonic weather types with mostly westerly flow). This alleviates the use of the two different reconstructions concerning consistency among the reconstructed variables.

The pre-processing steps of the gridded fields and observational data required additional reanalysis data. For detrending the gridded and observational data in the reference period, we used a zonal mean of the land-only daily (mean, maximum, minimum) 2 m temperature data from ERA-5 (Hersbach et al., 2020). For calculating a climatic offset between the reference period and the historical period to account for the warming since the pre-industrial period, we used the paleo-reanalysis ModERA





(Valler et al., 2024). This paleo-reanalysis is based on atmosphere-only general circulation model simulations and assimilates a variety of data types, such as early instrumental temperature and pressure data, documentary data, or tree-ring records.

# 3  Methods

## 3.1  Adjusting for temperature trends

Between 1971 and 2020, temperature trends have been significant, especially in the mountainous region of Switzerland (Me-
teoSwiss, 2025). To create a balanced analogue pool, we calculated the temperature trends using a linear regression of the zonal average of the daily mean, maximum, and minimum land-only ERA-5 2 m temperature. The resulting trends were removed separately for each temperature variable, using the corresponding daily values from ERA-5.

   Further, we had to consider the substantial increase in temperature from the start of our reconstruction period until the reference period. Therefore, we used the paleo-reanalysis ModERA (Valler et al., 2024) to calculate an estimate of the temperature
change between the historical period and the reference period. A monthly running offset over +/- 30 years between each historical year and the detrended average temperature for the reference year was calculated. This offset was added to the temperature observations to be able to compare measurements across the long periods. The same offset was removed from the temperature fields to create an accurate climatology of the past.

## 3.2  Analogue resampling method

We largely followed the analogue resampling method (ARM) from previous articles (Imfeld et al., 2023; Pappert et al., 2022; Pfister et al., 2020; Flückiger et al., 2017), however, with some differences due to available data sets. We created two new sets of resampled analogue days, one for the group of wind and relative humidity based on the COSMO-1 data with a reference period from 2016 to 2020, and one for the group of minimum/maximum temperature and relative sunshine duration with a reference period from 1971 to 2020. For both groups of analogue days, we used as a pre-selection the best weather type from
Pfister et al. (2024), and not as in the prior reconstruction a cumulation of weather types with a cumulative probability of 95 %. The following methodological differences were applied to the two data sets:

   – **Wind and relative humidity:** Because of the very small pool of analogue days from COSMO-1, we had to increase the seasonal window of days considered as possible analogue days to +/- 80 d. In total, this resulted in a window size of 160 d, which was then further reduced by selecting only days with the same weather types. Prior to calculating the analogue
170        days, all precipitation data were transformed to precipitation occurrence. The closest analogue days were then calculated using the Gower distance (Gower, 1971; Kuhn and Johnson, 2019). For the early networks, we used all available data, whereas for the period after 1864, we decided on a specific set of stations based on sensitivity analysis (see Fig. A2, blue boxes). In addition to pressure, temperature, and precipitation data, we also calculated indices accounting for the north-south and east-west pressure gradients. These additional indices are calculated based on the difference between
station pairs in north-south, respectively east-west direction, and are relevant to capture, for example, Foehn events.





- **Minimum and maximum temperature and relative sunshine duration:** The analogue pool for these variables with a reference period from 1971 to 2020 is much larger. We reduced the window size for analogue selection to +/- 45 d because this improved the results considerably. As in Imfeld et al. (2023), for the early period, the Gower distance was used, whereas for the later period, we used the RMSE. We used the full set of available observations in the early period, but only chose a specific selection of observations for the period after 1864, excluding precipitation. This can be justified based on the evaluation results.

All reconstructed variables, including their methods, reference period, and reference data sets, are listed in Tab. 1. Note that additional indices calculated from the meteorological variables have been presented in Imfeld et al. (2024a) and are not listed.

**Table 1.** Reconstructed variables and indices at the 1 km grid for Switzerland between 1763 and 2020. The table includes variables described in Imfeld et al. (2023) and in this article. ARM refers to the Analogue Resampling Method, EnKF to Ensemble Kalman Fitting, and QMAP to quantile mapping. Window refers to the window size used for the analog selection. PH describes the calculation of the reduced covariance matrix. The last column refers to the repository where the data is stored, which is Pangaea (Imfeld et al., 2022a) and BORIS Portal (Imfeld and Brönnimann, 2025).

| Description | Variable | Units | Method | Window | Reference data set | Reference period | Repository |
|---|---|---|---|---|---|---|---|
| Daily mean temperature | temp | $^\circ$C | ARM + EnKF | +/-60 d | MeteoSwiss (2021c) | 1961-2020 | PANGAEA |
| Daily mean precipitation | precip | mm/day | ARM + QMAP | +/-60 d | MeteoSwiss (2021a) | 1961-2020 | PANGAEA |
| Daily maximum temperature | tmax | $^\circ$C | ARM + EnKF | +/-45 d | MeteoSwiss (2021c) | 1971-2020 | BORIS Portal |
| Daily minimum temperature | tmin | $^\circ$C | ARM + EnKF | +/-45 d | MeteoSwiss (2021c) | 1971-2020 | BORIS Portal |
| Daily sunshine duration | srel | % | ARM | +/-45 d | MeteoSwiss (2021b) | 1971-2020 | BORIS Portal |
| Daily mean uv-wind | u10/v10 | m/s | ARM + EnKF | +/-80 d | COSMO-1 | 2016-2020 | BORIS Portal |
| Daily minimum relative humidity | rh | % | ARM | +/-80 d | COSMO-1 | 2016-2020 | BORIS Portal |
| Canadian Forest Fire Weather Index | fwi | unitless | derived | - | - | - | BORIS Portal |

### 3.3 Data assimilation for wind fields

We assimilated all available observations except for precipitation using an ensemble Kalman Fitting (EnKF) approach to further improve the resampled u- and v-wind fields. Ensemble Kalman fitting is an offline data assimilation approach, where the analysis is not passed to the next time step, i.e., every time step is handled individually (Bhend et al., 2012; Valler et al., 2022). Data assimilation tries to find an optimal representation of the true atmospheric state between the best guess of an atmospheric field (our analogue field) and the observations by minimizing a cost function (Franke et al., 2017). In the case of normally distributed errors, this cost function can be minimized with a Kalman filter. The best estimate of a true atmospheric state, referred to as the analysis $x^a$, is given by Eq. (1):

$$x^a = x^b + P^b H^T (H P^b H^T + R)^{-1}(y - H x^b) \tag{1}$$





where $x^b$ refers to the best estimate (the resampled analogue fields), $P^b$ is the model error covariance matrix, $H$ extracts the observations from the model space, and $R$ is the observation error covariance matrix. The second part on the right-hand side of the Eq. (1) $P^b H^T (H P^b H^T + R)$ is referred to as the Kalman gain $K$.

To account for a bias in the covariance analysis, we used the ensemble square root filter as proposed by Whitaker and Hamill (2002) and updated the ensemble mean and the anomaly from the ensemble mean individually, yielding two separate equations, Eq. (3) and (4).

$$\bar{x}^a = \bar{x}^b + K(\bar{y} - H\bar{x}^b) \tag{2}$$

$$x'^a = x'^b + \tilde{K}(y' - Hx'^b) \quad \text{with: } y' = 0 \tag{3}$$

The Kalman gain for the mean $K$ and anomaly $\tilde{K}$ were then calculated as follows.

$$K = P^b H^T (H P^b H^T + R)^{-1} \tag{4}$$

$$\tilde{K} = P^b H^T ((\sqrt{H P^b H^T + R})^{-1})^T \times (\sqrt{H P^b H^T + R} + \sqrt{R})^{-1} \tag{5}$$

We did not use a localization of the background error covariance matrix $P^b$ because this did not change the results in Imfeld et al. (2023). Because of the extremely small analogue pool in the reference period from 2016 to 2020, we did not calculate a covariance matrix based on the ensemble of 50 members available from the analogue resampling. For some of the reconstructed days, as few as 20 analogue days were available, considering our restrictions on weather types and seasons with a window size of +/- 80 days (see Sect. 3.2). Therefore, we tested different approaches to obtain an adequate representation of the background error covariance matrix, including (a) using only the available members, calculating constant covariance matrices for (b) each day of the year, (c) based on the weather type classification, and (d) using a blending of a fixed climatological covariance matrix ($P^{clim}$) and the covariance matrix of the target day based on the available members ($P^b$) as proposed in (Valler et al., 2024). Evaluations showed that using a blended covariance matrix led to the best results, though only by a small margin (Appendix, Fig. A2). The background error covariance matrix for this blending was defined as:

$$P^{blend} = 0.5 P^b + 0.5 P^{clim} \tag{6}$$

$P^{blend}$ was calculated using equally distributed weights of 0.5. Tests with different weights led to worse results, but no thorough testing has been performed because 0.5 seemed a reasonable choice. Whereas $P^b$ stems from the analogue days' ensemble considering the 20 best analogue days, $P^{clim}$ was calculated based on the weather type classification by randomly sampling 122 days from the same weather type (122 is the lowest number of occurrences of one weather types in 2016-2020). Due to the small sample size, we did not consider the seasons when calculating $P^{clim}$.



## 3.4 Data assimilation for maximum and minimum temperature

The data assimilation approach for maximum and minimum temperature follows the same equations as for the u- and v-wind field components. However, instead of pre-calculating a climatological covariance matrix and blending it, for every time step, the covariance matrix has been newly calculated based on the 50 best analogue days for each target day. For the period after 1864, we additionally used daily maximum and minimum values in the assimilation process. Maximum and minimum temperature were assimilated directly into the field after correcting for their monthly bias, calculated by comparing the observations with the closest grid cell in the reference period between 1971 and 2020. For the period before 1864, we only had daily mean temperature and could not assimilate this directly into the maximum and minimum fields, but we increased the state vector by adding the analogue days of the respective temperature observations. In the assimilation, the difference between the target day and the analogue day was calculated, and therefore, no bias correction was needed. The assimilation was performed on anomalies and the climatology, corrected by a climate offset (Sect. 3.1), was added at the end of the assimilation procedure.

## 3.5 Evaluation of reconstructed variables

The newly generated fields are evaluated based on two approaches. We performed a cross-validation by reconstructing all fields in the reference period, excluding 10 d around the target day for the ARM and the EnKF approach. We evaluated the daily data grouped into the four seasons and based on different metrics. As for application purposes, wind speed and wind direction are more often used than u- and v-wind components, we evaluated these variables. For wind speed, we calculated Pearson's correlation, a normalized root mean square error by dividing the root mean square error by the standard deviation of the observations (COSMO-1 wind speed), the mean squared error skill score (MSESS), and the mean bias. The MSESS is calculated using climatological mean values as a comparison. An MSESS for 1 indicates a perfect reconstruction, for an MSESS of zero, the skill of the reconstruction and climatology are equal, and for values below zero, the climatology outperforms the reconstruction (Jolliffe and Stephenson, 2012). For wind direction, we calculated the mean absolute error (MAE) based on degree values. For minimum and maximum temperature, relative humidity, and relative sunshine duration, we calculated the Pearson Correlation, the root mean square error (RMSE), the mean squared error skill score (MSESS), and the mean bias for each season. For the temperature variables, the evaluation was performed on anomalies with removed seasonality. As shown in Fig. 1a, the availability of observations changes largely throughout the years. In Imfeld et al. (2023), we evaluated the reconstruction approach for five different network setups. Here, we only showed the evaluation for a set-up corresponding to the station availability very early during our reconstruction period (corresponding to the year 1767, lowest amount of observations), for a network around 1819, and for the network as it is shown in Fig. 1 b and c. These networks are referred to as networks 1, 3, and 5 subsequently.

For the wind speed and temperature reconstructions, we further evaluated how well the extreme values were reconstructed by calculating for each grid cell the mean bias and RMSE for percentiles calculated based on the original data set. Additionally, we compared these grids to independent station observations, shown in Figure 1 (see asterisks in b and c), matching each





observation with its closest grid cell. Given Switzerland's complex topography, elevation differences between the 1 km grid cells and station locations can be substantial, leading to significant discrepancies depending on the variable. This must be considered when evaluating against station observations.

## 3.6 Canadian Forest Fire Weather Index

Using the reconstructed meteorological fields, we calculated the Canadian Forest Fire Weather Index (FWI), a widely used metric for fire danger studies worldwide (e.g., Van Wagner, 1987; Wotton et al., 2009; Vitolo et al., 2020). The FWI system comprises five sub-indices derived from the meteorological variables temperature, relative humidity, precipitation, and wind speed (e.g., Van Wagner, 1987), which build the final fire weather index (FWI). These sub-indices are the Drought Code (DC), the Drought Moisture Code (DMC), the Buildup Index (BUI), the Fine Fuel Moisture Code (FFMC), and the Initial Spread Index (ISI). The DMC, DC, and FFMC reflect varying levels of dryness at different depths of the soil and litter layers and over different timescales. The BUI combines the effects of DMC and DC, while the ISI integrates FFMC with wind speed to indicate the potential for fire spread. All sub-indices are open-ended, except for the FFMC, which ranges between 0 and 101, where 101 indicates that the fine fuel is easily ignitable. Based on the FWI, fire danger classes can be determined that are representative of their geographic region (see Kudláčková et al., 2024, for a global summary of danger classes). Originally, the FWI was calculated using instantaneous values at noon local time and 24 h precipitation accumulation (Van Wagner, 1987). However, our reconstructed data is only available at a daily resolution. Studies based on model simulations suggest that relative humidity at noon can be substituted with daily minimum relative humidity (e.g., Quilcaille et al., 2023). At the station level, we saw that using daily maximum temperature and daily minimum relative humidity produces FWI values comparable to those derived from noon observations (Appendix, Fig. A1). Consequently, our calculations of the fire weather indices are based on maximum temperature and minimum daily humidity.

## 4 Results

### 4.1 Wind fields

The cross-validation results for wind speed and direction reveal the best performance in winter and the weakest in summer (Fig. 2 and Fig. 3a-d). For the network as it was present around 1819, the Pearson correlation ranges between -0.07 and 0.9 in winter and -0.08 and 0.83 in summer (considering all grid cells), with lower values especially in the mountains and Foehn valleys. The normalized RMSE lies between 0.44 and 1.60 m/s in winter and 0.58 and 1.50 m/s in summer, while the MSESS across all seasons averages above 0.6, indicating that the reconstructed wind fields outperform climatological values. Across all seasons, the mean bias is slightly negative with values as low as -0.92 m/s, except for the Swiss Plateau, where slightly positive mean biases are seen (only up to 0.45 m/s). The MAE of the wind direction shows a less homogeneous spatial picture, with locally high values, particularly in DJF and SON, reaching up to 53.88° in DJF and 40.61° in SON. However, the average remains around 6.63° in DJF and 7.88° in SON.

**Figure 2.** Cross-validation of wind speed and wind direction for the period 2016 to 2020 and the four seasons DJF, MAM, JJA, and SON. a) Pearson correlation, b) normalized RMSE, c) mean squared error skill score, and d) mean bias of wind speed, and e) mean absolute error of wind direction. Evaluation metrics for wind direction are shown in degrees.

The differences between the available network sizes are significantly smaller than those observed for the precipitation and temperature reconstructions. While the smallest network, with only 11 observations (NW 1), produces considerably poorer reconstruction values, the results from the other networks are very similar (Fig. 3a-d). Network three delivers the best average results across the grid for the Pearson correlation coefficient and MSESS, but also exhibits the largest (though negative) average mean bias for spring, summer, and autumn. These findings suggest that adding more stations beyond a certain threshold does not further enhance reconstruction quality. For networks with more than 43 available records (NW 5), we tested various setups

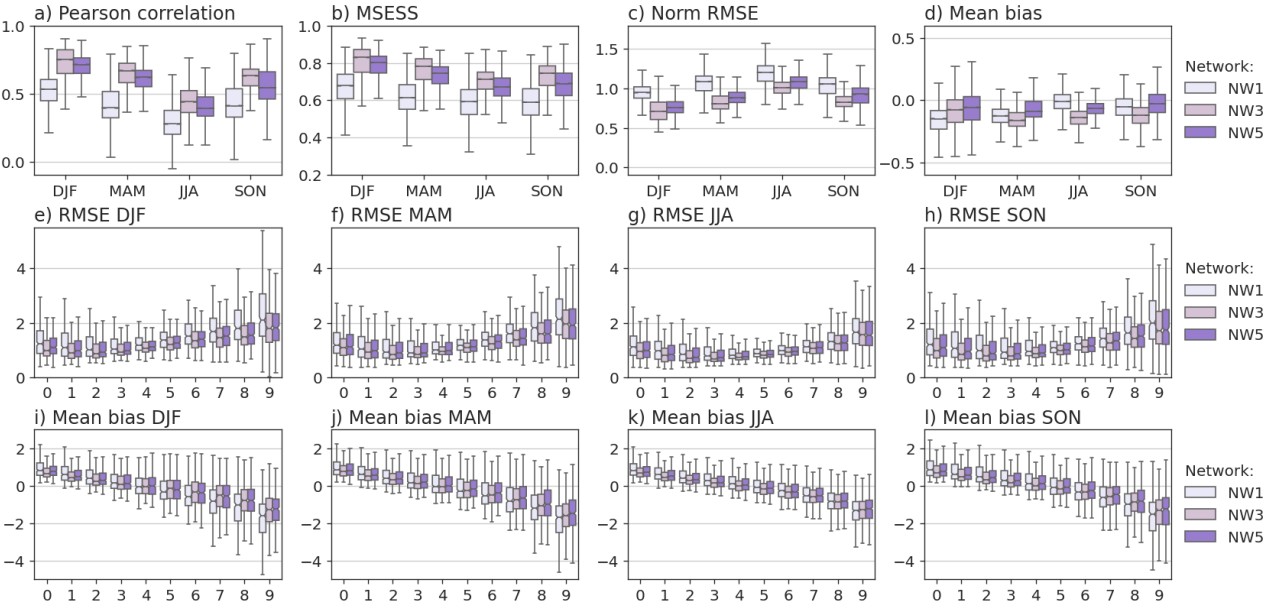

**Figure 3.** Cross-validation results of wind speed for different networks and seasons. a) Pearson correlation coefficient, b) MSESS, c) normalized RMSE, and d) mean bias of seasonal wind speed for five different networks, e-h) RMSE of percentiles wind speed for DJF, MAM, JJA, and SON for three networks, and i-l) mean bias of percentiles of wind speed for DJF, MAM, JJA, and SON. Percentiles are calculated based on the reference data set separately for each grid cell.

for the analogue resampling method (ARM) and data assimilation (Fig. A2). Selecting a specific subset of stations instead of all available stations increased the Pearson correlation by up to 0.1.

As expected, all seasons exhibit the largest RMSE values for the highest percentiles of wind speed, with the sparsest network consistently producing the largest errors (Fig. 3e-h). For the lower half of the percentiles, the RMSE remains relatively constant, though the spread of errors increases again for the smallest percentiles. This pattern suggests that very low wind speeds are more prone to overestimation, as reflected in the mean bias calculated by percentile (Fig. 3i-l). Across all seasons, the mean biases reveal an overestimation of low wind speeds and an underestimation of high wind speeds. This is likely due to the limited size of the analogue pool used for wind reconstruction, which affects the accurate representation of high wind speeds.

We further compared the wind reconstructions to independent measurements from the Saentis, Segl Maria, and Neuchâtel stations, with observations starting in 1882, 1864, and 1901, respectively (Fig. 4). Pearson correlation coefficients between the reconstructions and observations for the full overlapping period are 0.46 (Saentis), 0.25 (Segl Maria), and 0.53 (Neuchâtel), indicating rather low agreement as also seen in the comparison of time series and the scatterplots in Figure 4 (a-f). The highest correlations with station measurements are found for Geneva with a Pearson correlation coefficient of 0.56, which, however only starts measuring in 1958. Especially in the late 19th and early 20th, wind speed measurements had a lower measurement resolution (see lines of measurements at certain accuracies in Fig. 4b,d,f) and measurement problems occurred such as freezing

Earth System Science Data Discussions Open Access

of the anemometer on the Saentis at 2501 m a.s.l. (Hupfer, 2019). Also, the wind measurements have not been controlled for

quality and homogeneity, which can both affect the comparison with the reconstruction. Furthermore, a 1 km resolution grid

cannot fully capture the highly localized effects of wind in complex terrain, which further reduces agreement between the

reconstruction and observations, as shown by Kruyt et al. (2018) for the COSMO-1 model.

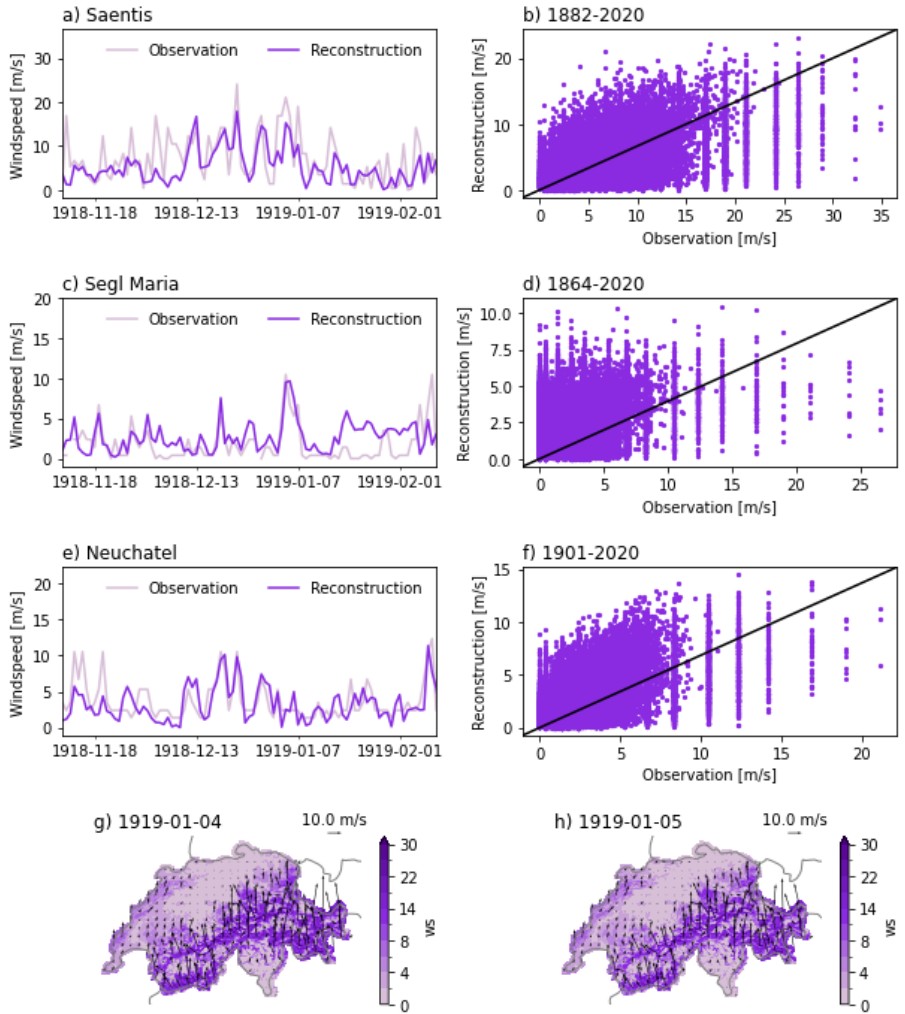

**Figure 4.** Comparison of reconstructed wind fields to independent observations. a) Time series for the location of Saentis compared to observations during a strong Foehn event from the 4th to 5th January 1919. b) Scatterplot of all measurements between 1882 and 2020. c-f) are the same as a and b) but for the stations of Segl Maria and Neuchâtel. g and h) 10 m wind speed (colour) and wind arrows for the strong Foehn event of 4th and 5th January 1919. Location of stations are shown in Fig. 1

From 4th to 5th January 1919, a strong Foehn storm occurred due to a record-low pressure north of the Alps, causing

substantial damage across Switzerland. This event has been classified as one of the eight extreme storms in Switzerland since



1856 (Stucki et al., 2014). For the time series, Segl Maria and Saentis, the reconstruction shows peak values around the 4th and 5th January (Fig. 4 a,c), but not for the station in Neuchâtel, which is not prone to strong Foeh winds. The strong Foehn winds are very pronounced throughout the Alps as seen in the maps of Switzerland (Fig. 4, e and f). However, this remains a rather qualitative assessment, and we cannot determine how realistically, for example, the spatial pattern of the wind field is
reconstructed.

## 4.2 Maximum and minimum temperature

The cross-validation results for maximum and minimum temperatures are presented for the network shown in Fig. 1b. For this network, we also incorporated the available maximum and minimum temperature observations. For comparison, the spatial analyses for the network around 1819 — when no maximum and minimum temperature observations were available — are
shown in Fig. A3.

For maximum temperature, cross-validation results show very high Pearson correlation coefficients across all seasons, with higher values especially north of the Alps, ranging between 0.77 and 0.99 for individual grid cells (Fig. 5a). In contrast, minimum temperature exhibits a more fragmented pattern, with the highest Pearson correlation coefficients concentrated around northern Switzerland and generally lower values south of the main Alpine ridge (Fig. 5d). For individual grid cells, the Pearson
correlation coefficients, however, still remain between 0.66 and 0.98. The RMSE shows very similar values between the two variables, with on average even slightly lower errors for minimum temperature (1.29 ° C) than for maximum temperature (1.33 ° C) and generally lower errors in the Swiss Plateau, where also on average more observations are available. Both maximum and minimum temperatures exhibit relatively few seasonal differences. In winter, the RMSE for minimum temperature reveals a distinct pattern, with the highest errors occurring in the region of La Brévine—one of the coldest areas in Switzerland due
to topographic and radiative effects. Additionally, RMSE values are larger in the higher elevations of the Canton of Ticino in southern Switzerland, likely due to the lack of observations at higher altitudes. Overall, biases range from -0.2° C to +0.2° C for maximum and minimum temperatures. The cross-validation for maximum temperature shows overall slightly better results than for minimum temperature, despite the availability of fewer observations for assimilation (10 series for maximum temperature and 14 for minimum temperature). This might be because maximum temperature values are spatially more homogeneous
and less influenced by local effects, such as lake effects and cold air pools.

The different networks show considerable differences, with the periods after 1864—when maximum and minimum temperatures can be assimilated—outperforming the earlier periods (Fig. 6a-d). For example, the network representing the period around 1819 (NW 3) has a mean Pearson correlation coefficient between 0.88 and 0.91, depending on the season for maximum temperature and between 0.81 and 0.86 for minimum temperature (see also Fig. A3 in Appendix). In contrast, the network at
the beginning of the reconstruction shows Pearson correlation coefficients of only between 0.75 and 0.88 on average per season for maximum temperature and between 0.73 and 0.76 on average for minimum temperature, which is considerably lower than network 5. Furthermore, both the minimum and maximum temperature in winter have much larger errors than in other seasons and networks with an RMSE of up to 3.1 ° C.


**Figure 5.** Cross-validation of maximum and minimum temperature for the period 1971 to 2020 and the four seasons DJF, MAM, JJA, and SON. a) Pearson correlation coefficient, b) RMSE, c) mean bias of maximum temperature, and d) Pearson correlation coefficient, e) RMSE, f) mean bias of minimum temperature. The maps show the cross-validation of the network as it is displayed in Fig. 1b (NW 5). All metrics are calculated based on temperature anomalies with removed seasonality.

The other metrics also show lower performance when no maximum or minimum temperature could be assimilated. The
earlier networks exhibit poorer results in summer and spring for minimum temperature, showing also a large spread in the
metrics. The mean biases are larger for the earlier networks, but remain within -0.22° C to +0.13° C for network 3.





The percentile evaluation has only been performed for network 5. It shows that the reconstruction has the largest RMSE for maximum temperature when estimating the highest temperature percentiles and for minimum temperature when estimating the lowest temperature percentiles (Fig. 6, e-h). For the average percentiles, errors for minimum and maximum temperatures are

relatively similar, though they are consistently slightly lower and exhibit less spread for maximum temperature. The mean bias evaluation indicates that the lowest temperature values are consistently overestimated in the reconstruction, while the highest values are consistently underestimated across all seasons, with the largest biases occurring in winter (Fig. 6, i-l).

It is furthermore relevant to consider whether the daily mean temperature always lies between the daily minimum and maximum temperature since we did not explicitly control for this in the reconstruction method. Our analogue fields for maximum

and minimum temperature already incorporate this, however, the daily mean temperature is based on different analogue days. Controlling the data sets for such inconsistencies shows that in the period from 1763 to 1863, for on average 2.5 % of the grid cells per year maximum temperature is lower than the daily mean temperature. The year 1812 has a record high with 4.2 % of the grid cells having lower maximum temperature values than daily mean temperature values. This high percentage stems from some individual days where the analogue days differed considerably. For minimum temperature, on average 1.7 % of

the grid cells per year show a higher value than the daily mean temperature. The largest fraction of days and grid cells above the daily mean temperature occurred in 1766, with 3.1 %. As soon as maximum and minimum temperatures were assimilated (starting in 1864), the percentage of too high/low values is reduced to on average 1.7 % for maximum temperature and 0.8 % for minimum temperature. However, the inconsistencies have a clear annual cycle, with most of the under- or overestimations happening in the winter season.

We further compared the reconstructions to measurements from Château-d'Oex and Segl Maria (see Fig. 1b for locations). Both locations are situated at the bottom of valleys in mountainous areas. While we used daily mean temperature during the assimilation, we did not include the maximum and minimum measurements from these stations in the reconstruction due to the short or incomplete nature of these series. For all four time series, the agreement between the closest grid cell of the reconstruction and the observation looks rather well (Fig. 7). For Château-d'Oex, the Pearson correlation coefficient is as high

as 0.92 for maximum temperature and 0.89 for minimum temperature in the period 1931-2020. For Segl Maria, the Pearson correlation coefficient is 0.88 for maximum temperature and 0.84 for minimum temperature in the period 1869-1970. RMSE for minimum and maximum temperature at both locations are between 1.82 and 2.15 ° C and mean biases are between -0.07° C and +0.03° C, all indicating that the reconstructions correspond well to the observations.

### 4.3 Relative sunshine duration

The relative sunshine duration reconstruction is derived from the analogue days used for maximum and minimum temperature reconstruction without additional post-processing. This analogue pool spans 50 years (1971–2020), providing a sufficiently large data set. Cross-validation results show a strong dependence on the observational network, with generally better performance for networks with a higher number of observations (Fig. 8a-c). Pearson correlation coefficients across all networks and seasons average above 0.4, though notable seasonal differences exist, with the best results in summer, followed by spring. The

MSESS values range on average between -0.21 in DJF (NW1) and 0.48 in JJA (NW5). For the network with the fewest obser-

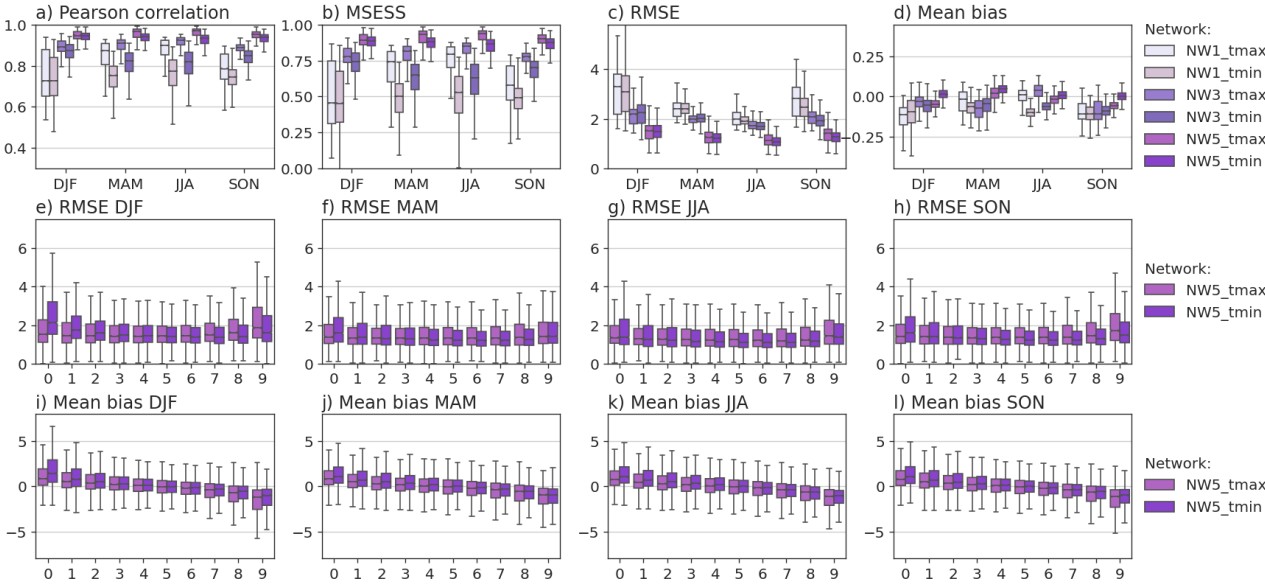

**Figure 6.** Cross-validation results of maximum and minimum temperature for different networks and seasons. a) Pearson correlation, b) MSESS, c) RMSE, and d) mean bias of seasonal wind speed for five different networks, e-h) RMSE of percentiles wind speed for DJF, MAM, JJA, and SON for three networks and i-l) mean bias of percentiles of wind speed for DJF, MAM, JJA, and SON. Percentiles are calculated based on the reference data set for each grid cell.

vations (NW1), the reconstruction of the relative sunshine duration is, therefore, largely outperformed by the climatology. For the network with the most observations, the reconstruction on average outperforms climatology as a reference for all seasons. However, it should be considered that for applications such as impact modeling, climatology may not be a suitable comparison because a climatological forecast lacks variance. The expected MSESS for predictions with the correct variance but no correlation is -1. The RMSE values are all very high, with average values between 20-40 %. Nevertheless, the mean biases are rather small, indicating a correct distribution as expected. The mean biases do not show a clear pattern based on seasons, but on networks. Networks 1 and 3 tend to overestimate the relative sunshine duration in all seasons, while network 5 tends to underestimate sunshine duration in winter. Note that for networks 1 and 3, all available data was used, whereas for network 5, only temperature data was used because this led to better results for the maximum and minimum temperature reconstruction.

## 4.4 Relative humidity

The relative humidity fields have been extracted from the COSMO-1 model output for the same analogue days used for the wind fields. Since we focused on reconstructing wind fields rather than relative humidity, we did not optimise the analogue selection for relative humidity. Accordingly, cross-validation results for minimum and daily mean relative humidity indicate limited skills, with daily mean values consistently performing better than minimum values (Fig. 8d-f). Pearson correlation coefficients range, on average, from 0.39 to 0.59, depending on the season and the network. The MSESS for daily mean and

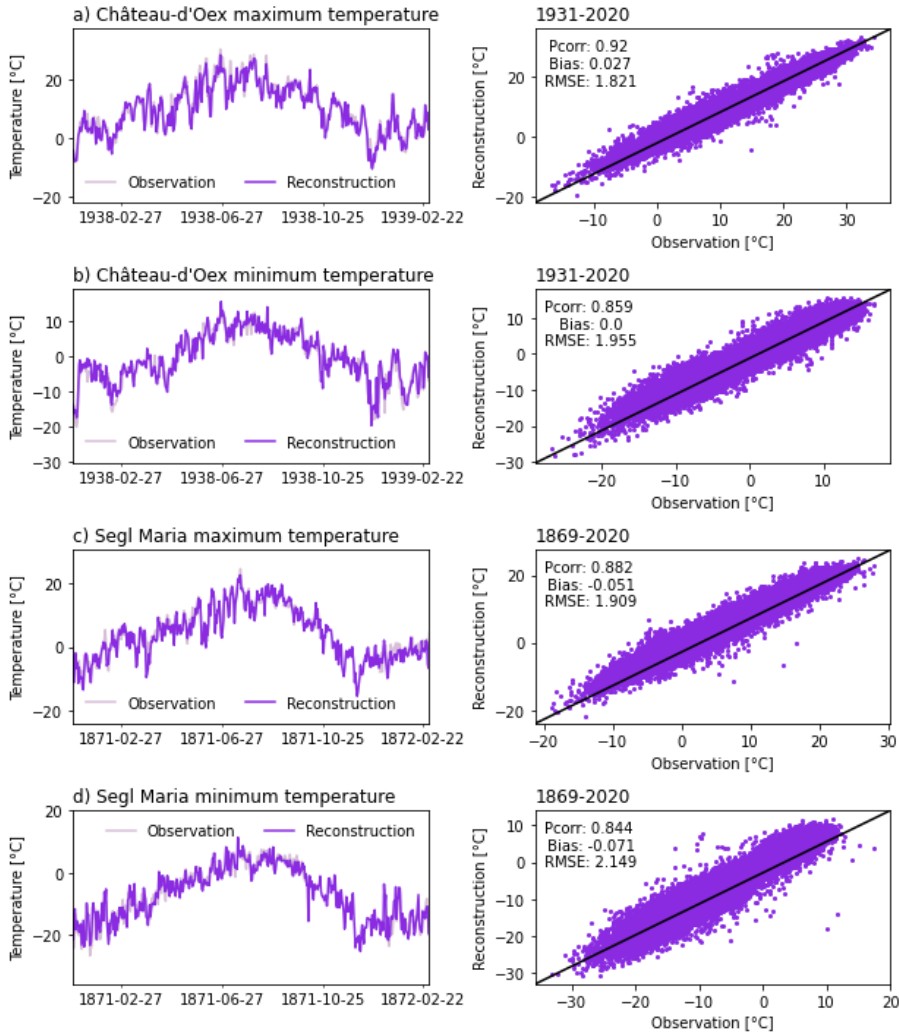

**Figure 7.** Comparison of reconstructed maximum and minimum temperature fields to independent observations. a and b) Time series for the location of Château-d'Oex compared to observations during one year of measurements and scatter plot for the entire period of measurements from 1931 to 2020. c and d) Time series for the location of Segl Maria compared to observations during one year of measurements and scatter plot for the entire period of measurements from 1869 to 2020 (with gaps). The time series have not been used for the reconstruction, but they have been used in creating the original gridded data set. The location of the independent stations is shown in Fig. 1.

minimum relative humidity ranges from -0.31 to 0.06, indicating that the reconstruction performs as good as the climatology or worse. These are not very convincing results, however, we optimised the analogue days for the wind reconstruction. Excluding, for example, many precipitation observations from the calculation of the analogue days could have deteriorated the relative humidity reconstruction. It should be considered that an advantage of our reconstructed variables is that they are physically largely consistent with each other as they stem from similar analogue days. Moreover, in comparison to using climatological


Data

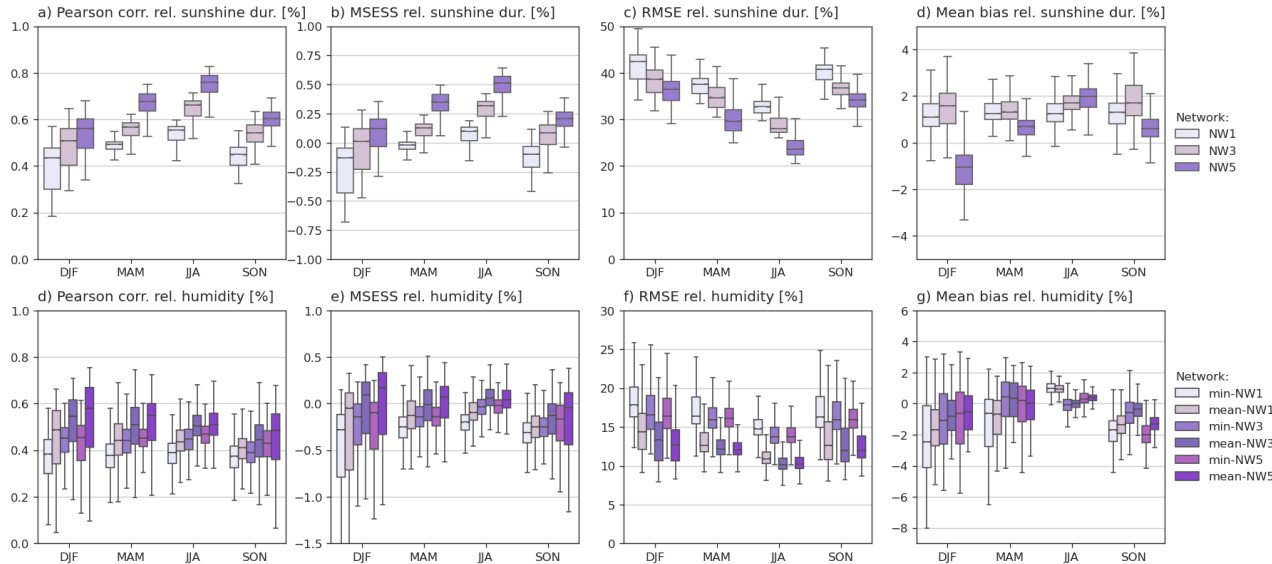

**Figure 8.** Cross-validation results for relative sunshine duration and relative humidity across different networks and seasons. a) Pearson correlation coefficients , b) MSESS, c) normalized RMSE, and d) mean bias of seasonal relative sunshine duration for five different networks. d) Pearson correlation, e) MSESS, f) RMSE, and fg mean bias for daily minimum and daily mean humidity considering the three networks. Relative sunshine duration is evaluated in the reference period from 1971 to 2020 and relative humidity is evaluated from 2016 to 2020.

mean values, our reconstruction captures the variance. The RMSE averages between 10–13 % for mean daily relative humidity and 14–18 % for minimum relative humidity. However, biases remain small, reaching at most -3 % on average. A small bias is expected because the values are sampled from the same distribution based on season and weather type. Both, the daily mean and minimum relative humidity are underestimated in all seasons, except for JJA. These relatively low scores are expected, as

no additional postprocessing was applied to the relative humidity fields, and the analogue pool for the selection of fields was very small. Nonetheless, we consider it important to provide these fields, since they are necessary for certain types of studies.

## 5 Long-term consistency of reconstructed fields

To qualitatively assess the long-term consistency of the reconstructed variables, we analyzed the 258-year field mean, smoothed using a 1-year running average. The wind speed reconstruction shows no substantial long-term changes. However, for a pro-

longed period around 1900, the field means do not exceed 2.25 m/s, suggesting a possible underestimation of wind speeds during that period. The temperature reconstructions agree very well with each other, which is not surprising since they all assimilate the same daily mean temperature data. Prominent features in the data set are, for example, the increase around the late 1940s, which is known as a warmer and drier period in the 20th century (Imfeld et al., 2022b). This same climate anomaly is also reflected in the increase of the relative sunshine duration around the late 1940s and in a slight decrease in the daily

mean relative humidity values. Relative humidity shows a slight decrease in values throughout the entire period. It is not clear



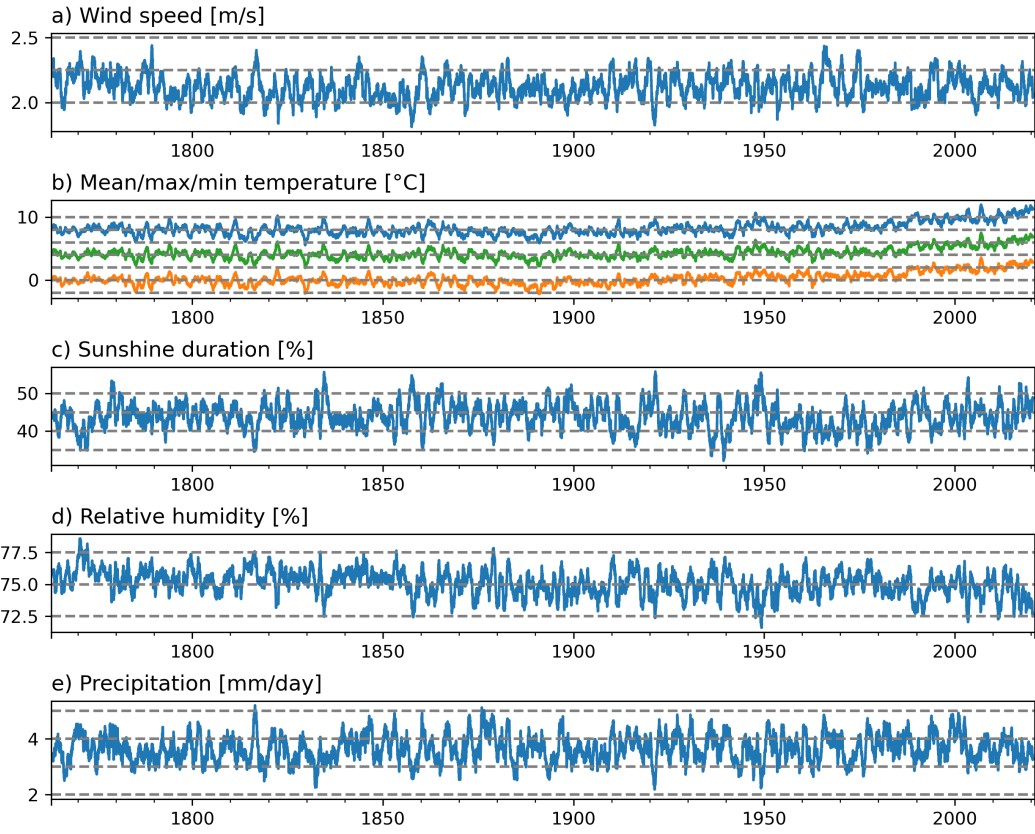

**Figure 9.** Long-term evolution of the reconstructed variables. a) daily wind speed [m/s], b) daily mean, maximum, and minimum temperature [° C], c) daily relative sunshine duration [%], d) mean daily relative humidity [%], and e) daily precipitation sums [mm/day] for comparison. The time series shows the area mean across the entire Switzerland, smoothed with a 1-year running mean.

whether this can be attributed to the reconstruction method and is therefore an artifact; however, studies on much shorter time scales suggest a decrease in relative humidity over Europe, for example, since the 1980 (e.g. Copernicus Climate Change Service, 2022). Certain patterns, such as lower values for sunshine duration, increased values for relative humidity, and increased values for precipitation between the 1830s and 1850s seem to be consistent in the different variables, despite the different analogue days as input.

## 6  Historical wild fire reconstructions

The reconstructed fire weather indices enable us to study the conditions that led to historical wild fires over the past centuries, while acknowledging that early data may be subject to uncertainty. As shown by the evaluation of the individual variables,

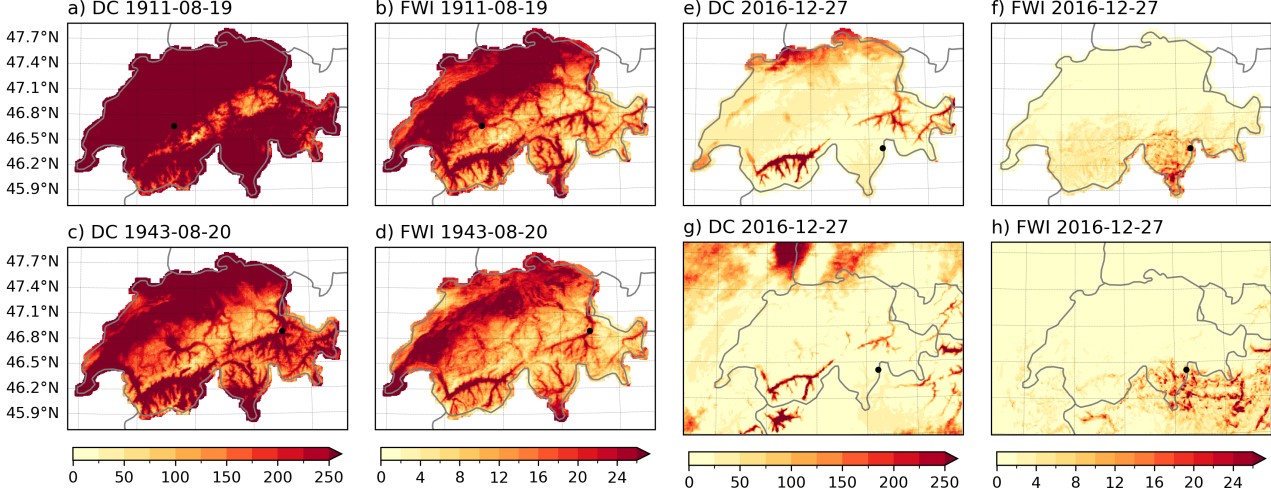

**Figure 10.** Drought code (DC) and fire weather index (FWI) for three forest fires. a and b) Forest fire in Simmenfluh, 1911, c and d) forest fire in August 1943 on the Calanda mountain slopes, e and f) forest fire in Misox in winter 2016, g and h) same but for COSMO-1. The approximate locations of the fires are shown as black asterisks in each panel.

the quality of the reconstruction depends strongly on the time period chosen, which directly affects the reliability of derived
indices such as the Canadian forest fire weather index.

Here, we examined three notable forest fires in Switzerland, one in August 1911, another in the summer of 1943, and a more recent fire in the winter of 2016, which allows for comparison with the COSMO-1 weather forecast model. On the 20th of August 1911, a forest fire broke out in the Canton of Bern, on the steep mountainous slope called the Simmenfluh (asterisks in Fig. 10 a and b) ignited by lightning from a passing storm, as stated in the newspaper "Die Berner Woche in Wort und Bild"
(), which was published at that time. Although the fire was extinguished within a few days, embers remained in the calcareous ground, reigniting it. The fire was only fully extinguished on the 22nd of September with the onset of heavy rainfall and led, in total, to damage of around 100 ha of burnt forest. The summer half year of 1911 was very dry and warm, as reflected in the high values of the DC (Fig. 10a). The FWI was variable across Switzerland on the 19th of August 1911 with high levels especially in the Swiss Plateau (Fig. 10 b).

Another significant fire occurred on the slopes of the Calanda massif above the city of Chur (asterisks in Fig. 10 c and d) following shooting exercises by the Swiss army on 20th August 1943. The fire grew rapidly due to the onset of Foehn winds and dry weather conditions, resulting in massive smoke plumes around the Calanda mountain (Bouchorikou, 2024; Fuchs, 2024). The fire was only fully extinguished after three days, causing significant damage to infrastructure and destroying 477 hectares of forest. For this event, the DC indicated particularly high values for the valleys around Chur, but also in the Swiss
Plateau and the southern valleys (see asterisks in Fig. 10c). The FWI, however, was less pronouncedly high across Switzerland, with only the Rhone Valley, the Swiss Plateau area, and the Rhine Valley showing values above 24 (Fig. 10d). The summer of 1943 was part of a series of relatively warm and dry summers in the 1940s (Imfeld et al., 2022a).



To compare the indices with those calculated from the model, we also considered two independent modern fires happening in the Moesa region (asterisks in Fig. 10 e and h) in December 2016, which led to the evacuation of two houses and caused

damage to around 120 ha of forest (Schildknecht, 2017; Fuchs, 2024). In contrast to the two historical fires, the DC and FWI values were considerably lower across Switzerland. Even though December 2016 has been exceptionally dry and with a very low snow-coverage (MeteoSwiss, 2017), the DC index does not show particularly high values for the Moesa region on 27th of December 2016 neither in the reconstruction nor the COSMO-model (see asterisks in Fig. 10). The FWI values were low across Switzerland, with a value of 2.72 on average in the reconstruction. FWI values were generally low across Switzerland,

averaging 2.72 in the reconstruction. However, in the village of Mesocco, located in the Moesa region, FWI values reached up to 18.7 on 28 December 2016 in the reconstruction (Fig. 10e–h), and the spatial pattern shows good agreement between the COSMO model and the reconstruction. In fact, the warning from the Federal Office of Environment, which is responsible for disseminating forest fire warnings, indicated a severe fire danger level on 23 December (level 3), and changed it to extreme (level 4) on 28 December 2016 when the fires were already burning. December 2016 has been exceptionally dry with very low

snow coverage in these areas, even though this is not seen from the DC index.

These short evaluations show that the data set seems suitable for studying certain climate impacts in the past, such as the occurrence of fire weather conditions in Switzerland.

## 7    Conclusions

This study expands high-resolution historical meteorological reconstructions by incorporating important variables, such as

wind, relative humidity, sunshine duration, and maximum and minimum temperature, alongside the existing reconstructions of daily mean temperature and daily precipitation sums. Spanning the period from 1763 to 2020, the data set represents an advancement for studying historical weather patterns in Switzerland, made possible through the integration of rescued early instrumental data, the analogue resampling method, and ensemble Kalman fitting.

Cross-validation results demonstrate convincing results for the temperature and wind variables for the period after 1864,

when in Switzerland a high-quality and dense observational network started to be set up. For this period, maximum and minimum temperature perform very well with Pearson correlation coefficients ranging between 0.81 and 0.91 and MSESS between 0.48 and 0.82. Wind speed also shows good correlations and MSESS values of 0.70 to 0.80 despite being derived from a very small analogue pool. Relative sunshine duration and relative humidity, however, exhibit moderate to low performance, not clearly outperforming climatological values. However, an advantage of our reconstruction is that the variables are largely

physically consistent with each other since they stem from the same or similar analogue days, and that they capture variance in the data set which is not given using climatological mean values.

The validation against observational data, especially for the temperature variables, further confirms the robustness of the reconstructed fields for this period, with correlations of up to 0.92 calculated on anomalies. Before 1864, when observations were sparser and of lower quality, maximum and minimum temperatures still perform considerably well, while wind speed

shows significantly worse performance, especially for the very sparse network in the 1760s. Nevertheless, the MSESS values

show that most of the reconstructed fields perform better than when using a climatological value, which can be helpful for certain applications. When working with this data, the limitations and the different skills especially for the variables of relative humidity and relative sunshine duration throughout the time periods have to be carefully considered. New methods, such as machine learning, and updated data products, such as time series of maximum and minimum temperature reaching further back

and in-depth homogenized pressure data, could help to substantially improve future versions of such weather reconstructions.

Despite these limitations, the data set described here is the first attempt to provide long-term, high-resolution gridded data for several key meteorological variables. The inclusion of these additional variables enables a broader range of applications, such as calculating the Canadian Forest Fire Weather Index, which provides valuable insights into historical fire weather and fire danger, as shown by the two examples of forest fires in 1911 and 1943 in Switzerland.

*Code availability.* The code is available at https://github.com/imfeldn/swiss-histmetgrids. We aim to publish a final referenced version at the end of the review process.

*Data availability.* The reconstruction data set is available at https://doi.org/10.48620/87086 in the Bern Open Data Repository (BORIS) (Imfeld and Brönnimann, 2025). It extends the already available data sets for daily mean temperature and daily precipitation sum available at https://doi.org/10.5194/cp-19-703-2023 (Imfeld et al., 2022a) and described in Imfeld et al. (2023). Indices calculated from daily temperature

and precipitation can be found in Imfeld et al. (2024a) and are described in Imfeld et al. (2024b). All variables and indices calculated in this article are listed in Tab. 1

**Appendix A**

**A1**



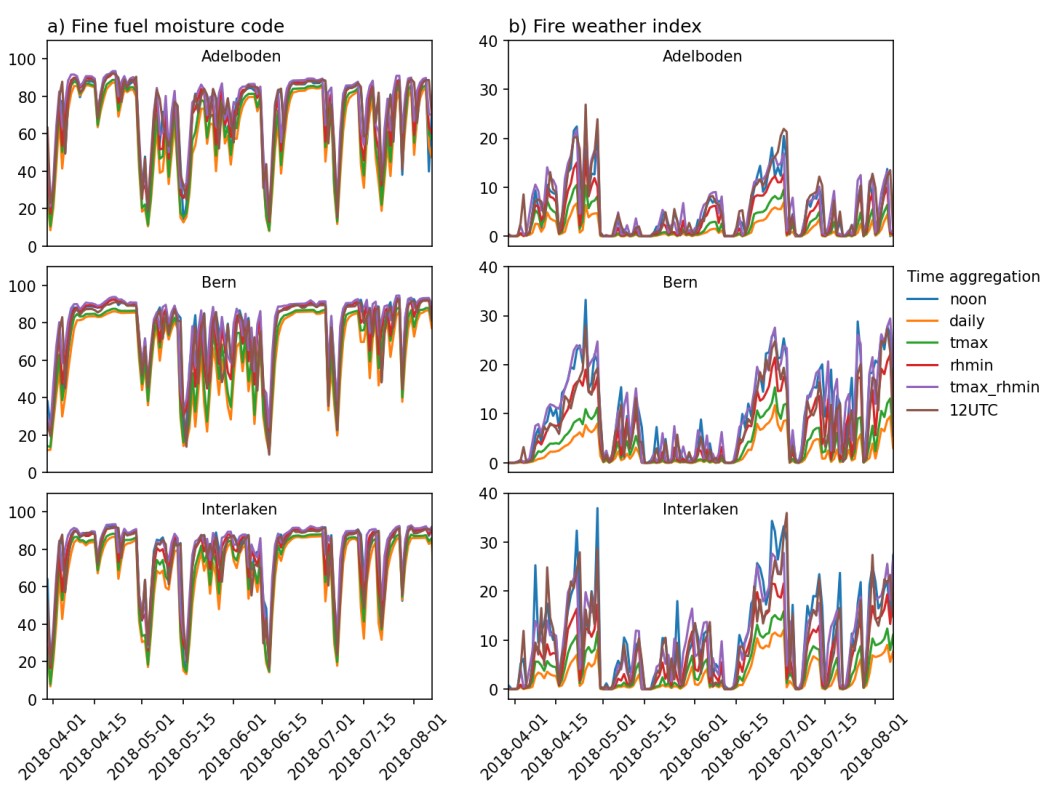

**Figure A1.** Comparison of FFMC and FWI calculated with different time aggregations for the weather stations in Bern, Adelboden, and Interlaken between April and August 2018. Noon refers to noon local time, daily refers to the daily mean aggregation between 0-24 o'clock for all variables except precipitation (6-6 aggregation), tmax and rhmin are maximum and minimum values between 0-24 of the respective variables. These evaluations show that using the daily maximum temperature and daily minimum relative humidity yields values similar to noon values.





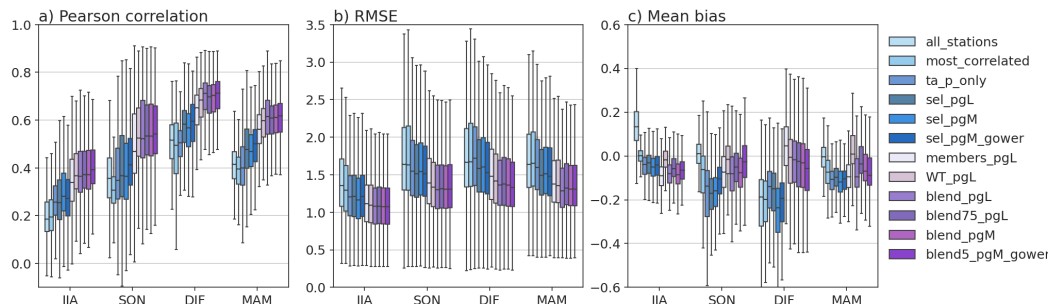

**Figure A2.** Comparison of evaluation results for the wind speed reconstruction depending on season. Blue colours represent evaluations for ARM only using different network set-ups, purple colours represent evaluation based on ensemble Kalman fitting, based on different calculations of the background error covariance matrix. Tab. A1 describes the different inputs.



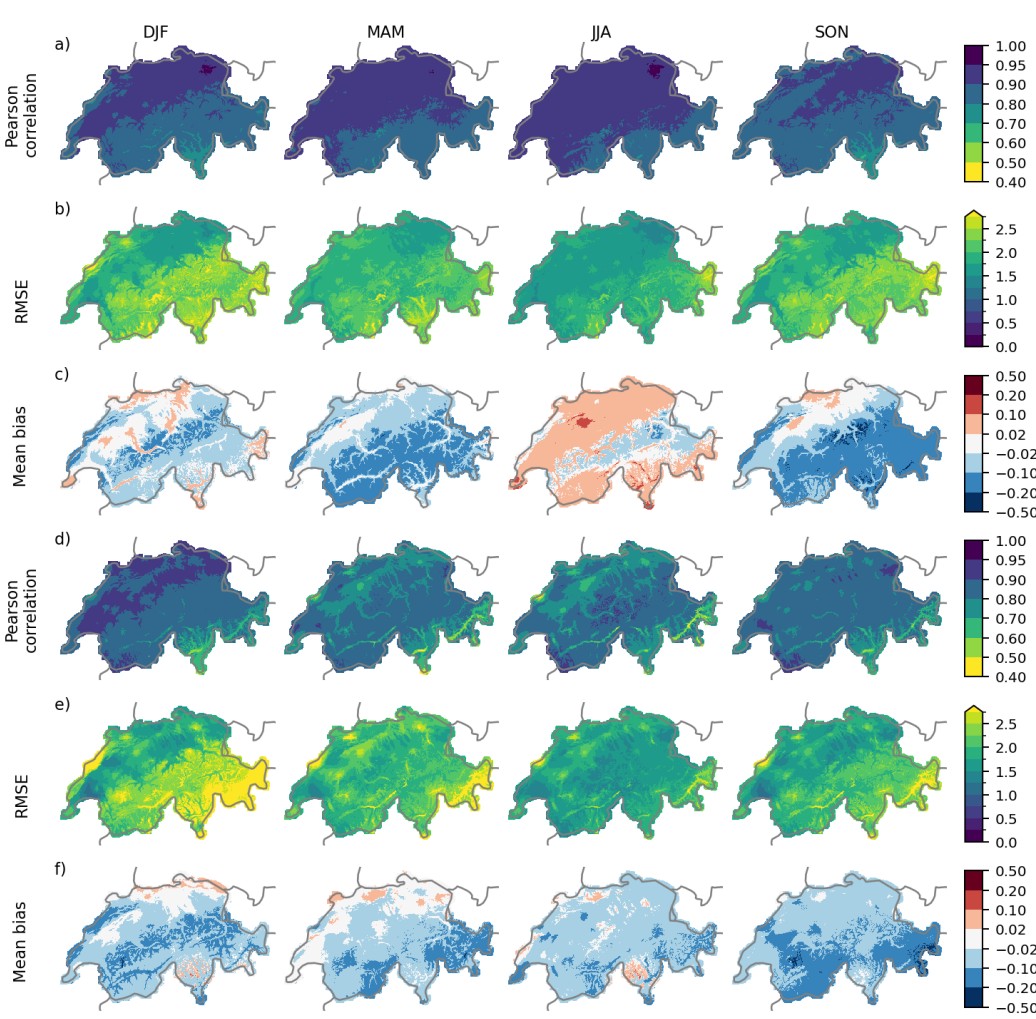

**Figure A3.** Cross-validation of maximum and minimum temperature in the period 1971 to 2020 for the four seasons DJF, MAM, JJA, and SON. a) Pearson correlation coefficient, b) RMSE, c) mean bias of maximum temperature, and d) Pearson correlation coefficient, e) RMSE, f) mean bias of minimum temperature. The maps show the cross-validation results of network 3. All metrics are calculated based on temperature data with removed seasonality.



**Table A1.** Description of different reconstruction set-ups for ARM and EnKF for reconstructing uv winds for the full network as it was available after 1934. The variables are daily mean temperature (ta), daily mean pressure (p), daily precipitation sum (rr), and daily precipitation occurrence (rr0). PH refers to the reduced covariance matrix.

| Name | Variables | Description | Distance measure | Reconstruction specifications |
|------|-----------|-------------|------------------|-------------------------------|
| all_stations | ta,p,rr,rr0 | all available stations | RMSE | ARM |
| most_correlated | ta,p,rr,rr0 | the 37 best correlated stations | RMSE | ARM |
| ta_p_only | ta,p | all temperature and pressure measurements | RMSE | ARM |
| sel_pgL | ta,p,rr0 | 51 stations with pressure gradient of Zurich-Lugano | RMSE | ARM |
| sel_pgM | ta,p,rr0 | 45 stations with pressure gradient of Zurich-Milano | RMSE | ARM |
| sel_pgM | ta,p,rr0 | 45 stations with pressure gradient of Zurich-Milano | Gower | ARM |
| members_pgL | ta,p,rr0 | as in sel_pgL with data assimilation | RMSE | EnKF with PH based on members |
| WT_pgL | ta,p,rr0 | as in sel_pgL with DA | RMSE | EnKF with PH based on weather types |
| blend_pgL | ta,p,rr0 | as in sel_pgL with DA | RMSE | EnKF with blended PH |
| blend75_pgL | ta,p,rr0 | as in sel_pgL with DA | RMSE | EnKF with blended PH (weight Pclim = 0.75) |
| blend_pgM | ta,p,rr0 | as in sel_pgM with DA | RMSE | EnKF with blended PH |
| blend_pgM_gower | ta,p,rr0 | as in sel_pgM with DA | Gower | EnKF with blended PH |

*Author contributions.* N.I. compiled the data, performed the reconstructions, wrote the manuscript, and produced all figures. S.B initiated
the study, supported developing the reconstruction method, and commented on the manuscript.

*Competing interests.* The contact author has declared that none of the authors has any competing interests.

*Acknowledgements.* This work was funded by the Wyss Academy for Nature, the Swiss National Science Foundation (project "WeaR", grant no. 188701), and by the European Commission through H2020 (ERC Grant PALAEO-RA 787574). The authors acknowledge the data provided by the projects "CHIMES" (SNF grant no. 169676) (Brugnara et al., 2020), "Long Swiss Meteorological series" funded by
MeteoSwiss through GCOS Switzerland (Brugnara et al., 2022), and "DigiHom" (Füllemann et al., 2011).



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
