# Peer review of "A daily gridded high-resolution meteorological data set for historical impact studies in Switzerland since 1763"

_Earth System Science Data, 2025_

## Author Comment (AC1)

Reviewer 1

The authors present a new reconstruction of daily sunshine duration, relative humidity, minimum and maximum temperature, and u- and v-wind with a 1x1 km resolution for Switzerland. The reconstruction covers the whole period since 1763, proposing thus a wide range of conditions that could be used as a baseline to analyse recent events or to estimate the impact of past changes on agriculture and fire development for instance. The methodology is similar to the one applied in a previous study. The skill of the reconstruction is evaluated in details to identify the interest of the approach but also the limitations. This is a very interesting product and it is well presented here. However, I would be happy to have a deeper justification of some of the choices and a longer discussion of the implications of those choices before the publication in the journal.

General comment.

My only general concern is about the consistency between the different variables. The authors mention (lines 469-471) that the 'an advantage of our reconstruction is that the variables are largely physically consistent with each other since they stem from the same or similar analogue days'. However, this consistency is ensured between some variables like relative humidy and wind speed but not for others. The daily mean temperature and daily precipitation comes from a previous study with a slightly different methodology. Wind and relative humidity come from a different pools of analogues compared to minimum and maximum temperature and relative sunshine duration. The possible issues are discussed for daily minimum and daily mean temperatures as inconsistencies can be obvious for those two variables but a wider discussion is needed. It is also not clear why different fields comes from different pools of analogs (for instance I guess COSMO is providing all the required variables ?) or why an updated reconstruction of daily mean temperature and daily precipitation was not produced here to be have a more consistent product instead of using the existing reconstruction.

We will add a new Chapter 5 to the manuscript covering the topics consistency and uncertainty (previously the chapter on long-term consistency). This chapter will include a more thorough discussion considering both topics. In order to address the consistency between variables we will conduct case studies in the reference periods of the data sets where we evaluate at how the different variables behave with respect to a physically consistent data set, such as the variables obtained from the COSMO model, and the different daily gridded products from MeteoSwiss.

Furthermore, we will add to the manuscript a more in-depth description why different variables come from different data sets. This is mainly related to the fact that larger analogue pools show better reconstruction results. The COSMO model is only used as an analogue pool, because there are no other promising datasets.

Specific comments

Line 16. I would mention that the two historical fires occurred in summer (to make clearer the difference with the winter contemporary wildfire).

We will include that the historical fires occurred in summer.

Line 36. It is mentioned 'mean and minimum relative humidity'. Is this correct ?

Yes, this is correct. We reconstructed these two variables because mean relative humidity is relevant for various applications, whereas we wanted to use and test minimum relative humidity for reconstructing the fire weather index.

Line 44. You should justify why you do not apply the data assimilation for those variables.

The units of these variables is percentage which has by definition upper and lower bounds. Since values, for example close to 100 % relative humidity or 0% relative sunshine duration occur regularly in the dataset, the data assimilation system would be needed to consider such non-gaussian distributions, e.g. by using logit-transformations. Furthermore, we do not have sufficient historical observations for humidity or relative sunshine duration and therefore, we could only assimilation daily mean temperature (and minimum or maximum temperature after 1864) onto the relative variable's fields. It would be interesting to further explore the methods to assimilate temperature data onto the relative humidity and sunshine duration fields, however this would require further in-depths studies on the exact implementation approach. This was beyond the scope of study in our case, which was mainly focusing on the reconstruction of minimum and maximum temperature and wind fields. However, we suggest it could be useful to test such methods in comparison to other methods (such as machine learning) to improve historical reconstructions of daily variables.

We will add a sentence discussing why we did not use data assimilation for the variables of relative sunshine duration and relative humidity.

Line 89. I do not follow to what corresponds the 361 values and later the 190 and 38 values. Which variables are selected and from which dataset ?

We will improve the explanation of what these values relate to by adding the following sentence:

*This additional quality control led to many values being flagged. When access to the "Wetterarchive" of MeteoSwiss was available, we checked whether these flagged values were digitization errors. We re-digitized 361 values where the error could be attributed to digitization. Furthermore, we re-estimated 190 values flagged by the spatial quality control to avoid losing relevant daily data. For example, when a possible digitization error was suspected but the original document was unavailable. In addition, 38 values were flagged as climatic outliers; these did not appear in the automated tests from the dataresqc R-package but were physically implausible based on the spatial tests. All flagged values were excluded from the reconstruction, except for those identified as daily repetitions.*

Lines 121-122. Is it possible to add a reference where this realistic representation is shown ?

We will add an additional reference that described the SrelD dataset.

Lines 126-128. Could you comment on the lower quality of the reconstruction using the data from COSMO-1 with ERA-5? I would have expected better results as a larger pool of analogs is available from the longer series. Additionally, it is mentioned 'using ERA-5 variables as predictors', I guess it is as boundary conditions for the COSMO-1 model.

In the cited study (Miralles et al. 2022), COSMO-1 wind fields were used to train a generative adversarial network (GAN) to predict hourly u and v winds based on ERA-5 variables as predictors. However, the evaluation results showed only limited ability to predict the wind

fields accurately. We performed our analogue reconstruction in the first place using the 60 years of daily wind fields from Miralles et al. 2022. However, cross-validation results for 2016-2020, comparing the analogue reconstructions using COSMO-1 and their wind fields separately, showed much better results when the reconstructions were based directly on COSMO-1 fields. Furthermore, whereas the reconstruction using COSMO-1 was able to capture (at least qualitatively) many known events, for example as show in Fig. 4, this was not the case for the reconstruction using the GAN-wind fields.

Whereas the study by Miralles et al. (2022) is certainly a very interesting piece of research on how to optimally implement an ML-model, in our case, it proved more useful to newly reconstruct the wind fields using the original COSMO-1 fields directly as an input into our analogue and data assimilation method.

Line 135. What is the impact of this choice ? Does it introduce inconsistencies compared to the previous reconstructions ?

Yes, it does introduce inconsistencies compared to our previous reconstructions. We will discuss this in an extended chapter 5 about consistencies in the data sets and we will in addition conduct an analysis of the consistency between variables based on two case studies, as already answered in the first comment.

Section 3.2 Could you explain how the number of analogs is selected (see for example lines 206-207)? You should also explain why two analogue pools were chosen and the potential consequences (see the general point above).

The lower number of analogues selected relates to the fact that based on our restrictions, there are not so many analogues available. To have a consistent number of analogues for every day, we used the lowest number of analogues available which is 20.

We will explain in more detail in the method section, why there are different analogue pools.

Line 248. Fig 1b and 1c correspond to year 1864 if I am right. I would write it explicitly for comparison with the two other dates (1767 and 1819).

We will adjust the headers of this figure to make more clear what exactly is shown. Furthermore, we will adjust Fig. 1 and add NW1 and NW3 as well.

Figure 2. It took me some time to see that panels a-d were for wind speed and panel e for wind direction. Is it possible to add the information directly on the plot ?

We will add this information to the Figure 2.

Line 284-285. This sentence is hard to follow. Does it make a reference to the previous study? In that case, I would expand the discussion to be more explicit.

We will add the reference to the previous study and reformulate the sentence:

*"The differences between the available network sizes are significantly smaller than those observed for the precipitation and temperature reconstructions presented in Imfeld et al. (2023). While the smallest network NW 1, with only 11 observations, shows considerably*

*lower skills for all four metrics, the results from NW 3 and NW 5 are very similar (Fig. 3a-d)."*

Figure 3. Are the I-l panels showing percentile (and in that case why only numbers between 0 and 9). Is it deciles instead ? (same for Figure 6).

We will adjust the caption of the figure, saying it contains deciles and we will adjust the x-axis of the panels e-l to 0.1-1.0.

Figure 5. As for Figure 2, would it be possible to put on the figure the variables shown ?

Yes, we will do this.

Figure 8. Typo in the caption 'fg' instead of 'g)'.

Thank you.

Line 430. Is a date missing between the parentheses ?

It should be the citation of the year when the newspaper was published. We will adjust this citation.

Lines 448-450. The same sentence is repeated twice.

Thanks, we'll delete it.

---

## Author Comment (AC2)

Reviewer 2:

The manuscript essentially presents an extension of long-term daily reconstruction data over Switzerland by six more variables, against previous work. Overall, I find the manuscript to be written clearly and a valuable add-on and dataset extension to the related previous studies. I therefore clearly rate it worth of publication in ESSD, upon revisions as suggested below. In particular, some parts are really in need of more detailed explanation, to make it easier for the reader to properly understand the study, and the aspects of quality and weaknesses of the data. While I rate all suggestions together to constitute a major revision, I don't think they are particularly difficult to implement, however. They mostly intend to aid improved clarity and readability of what basically is considered a decent paper. See the comments below.

**Major Comments**

**#1:** The different networks (NW 1, 3, and 5) used in the study are in need of a clearer and more consistent description and/or referencing. For example, an overview table including the NW's abbreviation, the number of stations, and the period a NW refers to, could be very helpful. In the text, I suggest to stick on using the NW abbreviations. The switching between the numbers and the descriptions makes it unduly (and without good cause) hard to keep in mind which network is which. Possibly also add a Fig similar to Fig. 1b and 1c for the other two NWs. Also, better add to the Fig captions on which NW the results shown are based on (see also the minor comment 1 related to the Fig captions).

We will consistently use the abbreviation NW1/3/5 in the text. Furthermore, we will add NW 1 and 3 as panels in Fig. 1 increasing the panel size of all maps. The respective periods and the number of observations will be added in the header of the panels in order to make it clear to what the different periods refer to.

**#2:** The description of the ARM appears a bit too minimalistic, while it should at least sufficiently convey the basic idea behind this method. While I do well understand that the description is kept short on purpose, since it is described in more detail in Imfeld et al. (2023), summarizing briefly also here how the analogue days are selected etc. would make it easier to understand how this very important step in the study works. In principle, this could also be explained aided by a related overview Figure.

We will add a paragraph in Chapter 3.2 explaining the ARM method in more detail, as it is done in the previous publication. However, we will not add an overview figure because the manuscript in already rather long with many figures. We hope the text fulfills the task of better explaining the method and refer to Imfeld et al. (2023) for an overview figure of the whole methodology since the general methodology stayed largely the same.

**#3:** Please state more clearly, which periods are compared to which in the evaluation. For example, I found the description in Imfeld et al. (2023) quite easier to understand, because it clearly stated that the reference period got compared to the historical period. In a way, the issue with the 'obscurities' re the different periods is similar to the one mentioned in major comment #1 above related to the NWs; it sometimes remains unclear to which periods a piece of discussion is referring. Hence please revise adequately and make sure that you use consistent descriptions/names of the periods.

All historical periods (described by different NWs) are compared to the reference period of the respective variable which is already shown in Tab. 1. Part of the confusion happens

probably because the reference periods are different depending on the variables. We will review the entire manuscript to clarify with which exact reference period an evaluation is performed. Furthermore, we will try to state more clearly why different reference period have been used for different variables.

**#4:** By always showing the results for Tmax and Tmin together, the Figs for these results get very crowded and it becomes hard to keep in mind which panel refers to which parameter. Please consider separating the results of Tmax from the ones of Tmin. If you want to keep them together (to show a direct comparison, which I don't see strongly needed for the science discussed), consider improving these Figs by adding subtitles that clearly state, where the results for Tmax are shown and where the ones for Tmin. Also, Figs 6 and 8 could benefit from reducing the number of colors in use. For example, consider sticking just to the 3 colors used for the 3 different NWs. Tmax and Tmin could then be distinguished by different color saturations, for example, or by getting the respective boxplots dashed. See also the minor comments 2 to 4 related to the Figs.

We will keep both variables, tmin and tmax, in Figure 6, however, we will change the color concept the way you suggested in order to make it easier to see which boxplot shows which variable (see example below). We will also adjust the colors in Fig. 8, and we will flip rows and columns of the figures, as mentioned in minor comment Nr. 2.

[Figure]

**#5:** Regarding the results of relative sunshine duration and humidity, the authors clearly state that their results are not convincing and the reconstruction performs (at best) only as good as

the climatology. I understand that the reconstruction focused on the wind variables and the setup was chosen accordingly, which is fine. However, if the reconstruction of relative humidity and relative sunshine duration are known to be of poor accuracy, how can they still be "important for certain types of studies"? Could these studies not simply use the climatologies instead? What is the added value of the reconstructed fields? Please either strengthen the scientific arguments or reconsider/tone down the way of inclusion of these two "lowest-quality" variables.

Please note that for relative sunshine duration, NW 5 clearly outperforms climatology, while NW3 largely outperforms climatology. These reconstructions can therefore be considered valuable as is.

Furthermore, note that while the MSESS for climatology as a no-knowledge prediction would by zero, the MSESS for (climatology plus) random variability as a no-knowledge prediction would yield a value of -1. For many applications, having the correct amount of variability matters. For these, -1 should be considered as the benchmark. On average, the reconstruction of both relative humidity and relative sunshine duration still range above this benchmark.

**#6:** Though the authors mention that "early data may be subject to uncertainty", the uncertainties are not explicitly quantified in the study. Could the authors provide at least a rough quantitative estimation of the associated uncertainties and how they develop over time? Or, at least, I strongly suggest to add a paragraph, where the uncertainties and their possible development over time are (semi-quantitatively) discussed.

We will change Chapter 5 (currently qualitatively discussing the long-term consistency) to include the topics consistency between variables and uncertainty.

Uncertainties in the data set stem from two main sources, once the uncertainty in the instrumental observations, especially for the early periods. The errors of individual time series have in more detail been discussed and estimated in the publication by Brugnara et al. (2022). This study can be seen as a reference on what types and magnitude of errors can be expected in the early instrumental series. However, all series show different error magnitudes which are not consistent over time. The series of publications in Brönnimann (2020) provides for the Swiss series a further idea on the quality and uncertainties in the observations.

Furthermore, uncertainty arises from the reconstruction method itself. While the analogue resampling approach provides an ensemble that could be used to estimate uncertainty, the way we implemented the analogue approach only provides us with an ordered ensemble. The first member of the ensemble can be considerably more close to the true state of the atmosphere than the 10$^{th}$ members, depending on the value of the distance function. The more suitable approach to estimate uncertainty would be to disturb observations and re-estimate the best analogue based on the disturbed observations, which we did not apply here.

In order to still give an idea of the uncertainty in the data set, we will use the case study of consistency between variables to also show the spread of the ordered ensemble. Note that in Imfeld et al. (2023) we already presented the 10 best members of the daily mean temperature reconstruction, however, after calculating a phenological index. This reconstruction shows barely any differences between the best 10 members.

**Minor comments**

1: On the Fig. captions: some of the captions are done a bit 'lazy', missing important information (e.g., which NW used) and are sometimes not correct (e.g., "[…] for *five* different networks […]" in Fig. 8, or saying "[…] shown as *black asterisks* […]" in Fig. 10, though dots are used). Please correct, and concisely complete the description of what is actually visible (where needed).

We will make the captions more comprehensive and include more information.

2: Specifically on the Figs with boxplots: These plots are quite 'crowded' and hence hard to grasp. Please think, for example, to somehow increase the horizontal space available for each plot (maybe by flipping the rows and columns). And/or consider to restrict to the most important boxplots really discussed in the text.

For Figures 3, 4, 6, and 8, we will transpose the layout by swapping rows and columns to make better use of the available horizontal space. In addition, we will adopt a revised color scheme to ensure the boxplots are more easily distinguishable.

3: Specifically on Fig. 1: The maps are somewhat hard to read. Consider, for example, the following changes:

- Rearrange the individual panels. Panel a could be on top and the maps below. And if you stretch panel a slightly, the horizontal space available for the maps increases, enhancing their readability.
- Use colors that can be distinguished more easily. In particular, the Tmean and Tmax/Tmin colors do appear not suitable in the maps because they look very similar.
- The asterisks are quite hard to see. Consider using a symbol that is more clearly distinguishable from the dots. Also, the black color appears too similar to the purple used for pressure.

We will rearrange this figure including NW 1 and 3. We use one row for the evolution of the network (a), and then 2x2 panels for the four examples of the networks for which later in the article the evaluation results are shown. We will change the colors to increase the visibility of the different variables.

4: Specifically on Fig. 4: The maps in panels g and h appear really too small, and the wind arrows are hardly visible in fact. Consider moving the maps to a separate bit-larger Fig (well possible to have 4a-f separate).

We will change the orientation of Figure 4 in order to increase the available space for panel g and h (see example below).

[Figure]

5: On the linking of dates/periods in the text to Figs: the text at several places refers to certain dates/periods and discusses what can be seen in the Figs around these dates/periods. Consider, as applicable, marking these dates with vertical (e.g., dashed) lines in the Figs and adding, for periods, a shaded gray area in the Figs. This would help with finding the features referred to in the text more quickly.

We will add a shading to the respective periods in the figure.

6: On occasional split of numbers and units in physical quantities: Better use "unbreakable" spaces between numbers and units; ensures that this is always well legible even if line breaks interfere.

This will be provided in the final version of the article.

7: In line 1: Typo in this line, "… data is needed to …" should read "… data are needed to …" (data are a plurality of individual elements, hence plural form is right; please check, and rectify as needed, throughout the text).

Thank you, we will correct this.

8: In lines 14-16, and at end of Abstract: please better embed the meaning of "... The two historical fires ..." (it is not clear from text context before, how the case "two fires" suddenly pops up here and which these are). And: at the end of an ESSD paper abstract it is good practice to explicitly note the data availability (including DOI) in a closing sentence.

We adjust the sentence to better include the fires.

*"Furthermore, we explored the potential of the extended reconstructions by evaluating two historical and one contemporary wildfire events in Switzerland using the widely used Canadian Forest Fire Weather Index (FWI). The two historical fires that occurred in the summer were associated with a notably high fire danger in the reconstruction."*

And we added the following sentence at the end of the abstract:

*"The datasets are freely available at the BORIS repository of the University of Bern: ttps://doi.org/10.48620/87086 (Imfeld and Brönnimann, 2025)."*

9: In line 47: Typo in this line, "… to compare it two …" should read "… to compare it to …"

Thank you, we will correct this sentence.

10: In line 430: reference to the cited newspaper seems missing (just a blank "()" visible). Maybe you also might meanwhile have a more scientific reference? (but if the news source is properly cited, ok also in this case).

We will correct the citation. Unfortunately, there is no more scientific reference.

11: Finally, on data availability (DOI in line 487): looks data are orderly available and carefully prepared, and reasonably simple to download via the .zip files; well in line with what the standards should be for an ESSD paper. These few suggestions, nevertheless:

a. fire weather indices (fwi) would be, in my view, more readily aligned for use if the six files were stratified by index, instead of chunked into subperiods of time. That is, one full _1763-2020.zip file per index, and six files in this way (if keeping with the apparent goal to safeguard max. size to no more than ~15 GB per file).

Usually, the different fire weather indices are considered in combination. Therefore, we think storing the indices in packages of different subperiods seems to us more helpful than separating the files by index.

b. the "readme.txt" is basically fine but it could have simply dropped all those subinfo-points that are anyway unused (but well covered via the paper). For example, under "Methodological Information", all but the first point can be dropped since the unfilled stuff makes it look 'unfinished' etc. And: the umlaut characters (e.g., "ö" in Brönnimann) don't correctly unfold in an EN-based viewer/at least not in mine (perhaps resort to pure ascii, like "oe" and so).

We will change the umlaut character to oe.

References:

Brönnimann, S. (Ed.): Swiss Early Instrumental Meteorological Series, Geographica Bernensia G96, Bern, https://boris.unibe.ch/173023/1/G96.pdf (last access: 24 March 2023), 2020.

Brugnara, Y., Hari, C., Pfister, L., Valler, V., and Brönnimann, S.: Pre-industrial temperature variability on the Swiss Plateau derived from the instrumental daily series of Bern and Zurich, Clim. Past, 18, 2357–2379, https://doi.org/10.5194/cp-18-2357-2022, 2022.

Imfeld, N., Hufkens, K., and Brönnimann, S.: Extreme springs in Switzerland since 1763 in climate and phenological indices, Climate of the Past, 20, 659–682, https://doi.org/10.5194/cp-20-659-2024, 2024